# Identification of Unique microRNA Profiles in Different Types of Idiopathic Inflammatory Myopathy

**DOI:** 10.3390/cells12172198

**Published:** 2023-09-02

**Authors:** Sandra Muñoz-Braceras, Iago Pinal-Fernandez, Maria Casal-Dominguez, Katherine Pak, José César Milisenda, Shajia Lu, Massimo Gadina, Faiza Naz, Gustavo Gutierrez-Cruz, Stefania Dell’Orso, Jiram Torres-Ruiz, Josep Maria Grau-Junyent, Albert Selva-O’Callaghan, Julie J. Paik, Jemima Albayda, Lisa Christopher-Stine, Thomas E. Lloyd, Andrea M. Corse, Andrew L. Mammen

**Affiliations:** 1Muscle Disease Unit, National Institute of Arthritis and Musculoskeletal and Skin Diseases, National Institutes of Health, Bethesda, MD 20892, USA; iago.pinalfernandez@nih.gov (I.P.-F.); maria.casal-dominguez@nih.gov (M.C.-D.); katherine.pak@nih.gov (K.P.); jcmilise@clinic.cat (J.C.M.); josetorresruiz85@gmail.com (J.T.-R.); 2Department of Neurology, Johns Hopkins University School of Medicine, Baltimore, MD 21205, USA; lchrist4@jhmi.edu (L.C.-S.); tlloyd4@jhmi.edu (T.E.L.); acorse@jhmi.edu (A.M.C.); 3Muscle Research Unit, Internal Medicine Service, Hospital Clinic de Barcelona, 08036 Barcelona, Spain; jmgrau@clinic.cat; 4CIBERER, IDIBAPS, University of Barcelona, 08036 Barcelona, Spain; 5Translational Immunology Section, National Institute of Arthritis and Musculoskeletal and Skin Diseases, National Institutes of Health, Bethesda, MD 20892, USA; lushajia@ep.niams.nih.gov (S.L.); gadinama@mail.nih.gov (M.G.); 6Genomic Technology Section, National Institute of Arthritis and Musculoskeletal and Skin Diseases, National Institutes of Health, Bethesda, MD 20892, USA; faiza.naz@nih.gov (F.N.); gutierrg@exchange.nih.gov (G.G.-C.);; 7Department of Immunology and Rheumatology, Instituto Nacional de Ciencias Médicas y Nutrición Salvador Zubirán, Mexico City 14080, Mexico; 8Systemic Autoimmune Diseases Unit, Vall d’Hebron General Hospital, Universitat Autònoma de Barcelona, 08035 Barcelona, Spain; albert.selva@vallhebron.cat; 9Division of Rheumatology, Department of Medicine, Johns Hopkins University School of Medicine, Baltimore, MD 21205, USA; jpaik1@jhmi.edu (J.J.P.); jalbayd1@jhmi.edu (J.A.)

**Keywords:** inflammatory myopathies, myositis, miRNA, NanoString, nCounter

## Abstract

Dermatomyositis (DM), antisynthetase syndrome (AS), immune-mediated necrotizing myopathy (IMNM), and inclusion body myositis (IBM) are four major types of idiopathic inflammatory myopathy (IIM). Muscle biopsies from each type of IIM have unique transcriptomic profiles. MicroRNAs (miRNAs) target messenger RNAs (mRNAs), thereby regulating their expression and modulating transcriptomic profiles. In this study, 18 DM, 12 IMNM, 6 AS, 6 IBM, and 6 histologically normal muscle biopsies underwent miRNA profiling using the NanoString nCounter system. Eleven miRNAs were exclusively differentially expressed in DM compared to controls, seven miRNAs were only differentially expressed in AS, and nine miRNAs were specifically upregulated in IBM. No differentially expressed miRNAs were identified in IMNM. We also analyzed miRNA-mRNA associations to identify putative targets of differentially expressed miRNAs. In DM and AS, these were predominantly related to inflammation and cell cycle progression. Moreover, our analysis showed an association between miR-30a-3p, miR-30e-3p, and miR-199b-5p downregulation in DM and the upregulation of target genes induced by type I interferon. In conclusion, we show that muscle biopsies from DM, AS, and IBM patients have unique miRNA signatures and that these miRNAs might play a role in regulating the expression of genes known to be involved in IIM pathogenesis.

## 1. Introduction

Idiopathic inflammatory myopathies (IIM) are a heterogeneous family of diseases that includes four major types as follows: dermatomyositis (DM), antisynthetase syndrome (AS), immune-mediated necrotizing myopathy (IMNM), and inclusion body myositis (IBM) [1]. Myositis-specific autoantibodies (MSA) are found in up to 60% of IIM patients and define unique phenotypic subtypes.

Muscle biopsies from each major type of IIM have distinctive pathological features as well as unique transcriptomic signatures. Indeed, because each type of IIM has a disease-specific gene expression pattern, machine-learning algorithms can classify muscle biopsies from IIM patients with greater than 90% accuracy using only transcriptomic data as an input [2].

MicroRNAs (miRNAs) are small non-coding RNAs that negatively regulate the expression of target messenger RNAs (mRNAs) by binding to their sequence and leading to the inhibition of their translation or to their degradation [3]. Gene expression regulation by miRNAs has been widely implicated in multiple biological processes, including inflammation and muscle regeneration [4,5,6], and altered miRNA expression has been reported in several diseases [7,8]. We hypothesize that the differential expression of miRNAs in IIM muscle may play a role in shaping the unique transcriptomic profile found in each type of IIM. Although prior studies have found differentially expressed miRNAs in muscle biopsies from different types of IIM, these did not include representative samples from each of the major types and MSA-defined subtypes of IIM.

In this study, we utilized the NanoString nCounter system to quantify the expression of ~800 miRNAs in muscle biopsies from patients diagnosed with MSA-defined DM, AS, and IMNM as well as IBM. This revealed that each major type of IIM has a distinctive miRNA profile that may play a role in shaping the transcriptomic signature of each disease. Additionally, we combined miRNA data with mRNA sequencing data obtained from the same samples. This has provided an integrated view of the coordinated changes of miRNA and mRNA expression that occur in IIM and that could have relevant pathophysiological roles in these diseases.

## 2. Materials and Methods

### 2.1. Subjects and Sample Preparation

Forty-two samples from patients with IIM and 6 controls were randomly selected for miRNA profiling from a collection of RNA samples from muscle biopsies used in earlier studies [2,9,10]. IIM patients had been classified as DM, IMNM, or AS, based on the presence of MSAs defined for these IIM types, according to Casal and Pinal criteria [11], or as IBM, based on the Lloyd diagnostic criteria for this subgroup [12]. IIM patients included 18 DM (5 anti-NXP2^+^, 5 anti-Mi2^+^, 2 anti-MDA5^+^, and 6 anti-TIF1g^+^), 12 IMNM (6 anti-HMGCR^+^ and 6 anti-SPR^+^), 6 AS (anti-Jo1^+^), and 6 IBM. The muscle biopsies classified as controls were obtained from patients presenting with muscle pain or cramping but without objective muscle weakness. In five patients, CK levels were normal. One patient had a moderately elevated CK at 397 U/L. None of the control muscle biopsies had inflammatory cell infiltrates, perifascicular atrophy, rimmed vacuoles, myofiber necrosis, or other features associated with IIM. The histological features and clinical information associated with each sample are summarized in Appendix A. Written informed consent was obtained from each subject as a requirement for their inclusion in the study, which was approved by Institutional Review Boards in the three centers enrolling subjects (National Institutes of Health, Johns Hopkins Hospital, and Clinic Hospital in Barcelona).

For sample preparation, muscle biopsies were performed, the tissue was flash frozen, and total RNA was extracted using TRIzol reagent (Thermo Fisher Scientific, Waltham, MA, USA, 15596018). RNA integrity was assessed with the High Sensitivity RNA ScreenTape assay (Agilent, Santa Clara, CA, USA, 5067–5579), and RNA concentration was measured using the Qubit RNA high-sensitivity assay (Thermo Fisher Scientific, Waltham, MA, USA, Q32852).

### 2.2. NanoString miRNA Assay and Analysis of nCounter Data

Total RNA was processed to analyze miRNA expression according to the nCounter human v3 miRNA assay (NanoString, Seattle, WA, USA, CSO-MIR3-12) following the manufacturer’s protocols. Counts for the miRNAs included in the panel were obtained using the nCounter MAX digital analyzer and the nSolver 4.0 analysis software. Quality control of the assay, normalization of the raw counts, and differential expression analysis of miRNAs were performed following an approach benchmarked by Bhattacharya et al. [13], with some modifications. These included the filtering out of miRNAs based on their counts relative to the negative controls or to a possible cross-detection of the reporter probes. To adapt the analysis to miRNA data, another modification was the selection of a subset of miRNAs to be used as reference probes for normalization. Apart from these adjustments, which are described in detail in the Appendix A, the analysis of NanoString counts was performed as indicated by Bhattacharya et al. [13]; technical variation was estimated using the RUVSeq package [14] (v.1.28.0) and considered during differential expression analysis, which was performed using the DESeq2 package [15] (v.1.34.0), and during normalization of counts to log-scaled values, which were used for graphical representation. The Benjamini–Hochberg correction was used to adjust for multiple comparisons, and an adjusted *p*-value < 0.05 and an absolute fold change >1.5 were selected as thresholds to consider a miRNA as differentially expressed. Differential expression analysis results were visualized using the EnhancedVolcano package (v.1.12.0).

### 2.3. Library Preparation and Analysis of mRNA Sequencing Data

Library preparation and sequencing of bulk RNA were performed as previously described [2]. Briefly, mRNAs were enriched from total RNA, and libraries were prepared using the TruSeq Stranded mRNA Library Prep Kit (Illumina, San Diego, CA, USA, RS-122-2103). Sequencing was performed on Illumina HiSeq2500 or 3000 platforms. Then, following demultiplexing of reads with bcl2fasq (v.2.20.0), FASTQ files for each sample were preprocessed using fastp [16] (v.0.21.0). Counts for the transcripts annotated in the human genome (GRCh37) were obtained using Salmon [17] (v.1.5.2). To review the output from these tools, reports were generated using multiqc [18] (v.1.11). Differential expression analysis of mRNAs was performed on raw counts using DESeq2. Adjustment for multiple comparisons was performed with the Benjamini–Hochberg correction, and the threshold for differential expression was set to an adjusted *p*-value < 0.05. For graphical representation, normalization of mRNA expression was obtained using the trimmed mean of M-values (TMM) method [19] from edgeR (v.3.34.1), and the values were log_2_ transformed.

### 2.4. Analysis of miRNA—mRNA Associations

To investigate associations between the expression of miRNAs and between the expression of miRNAs and mRNAs, we calculated Spearman’s rank correlation coefficients (r_s_) and the associated *p*-values using the normalized expression data. Search for candidate miRNAs targets was performed using the multiMiR package [20] (v.1.16.0). Pathway enrichment of potential targets was assessed using the enricher function from clusterProfiler (v.4.2.2) [21] and the Molecular Signatures Database (MSigDB) hallmark gene set collection [22]. The dot plot function from clusterProfiler was used to picture enrichment results.

## 3. Results

### 3.1. Unique miRNA Profiles in Each Type of IIM

A total of 18 DM, 12 IMNM, 6 AS, 6 IBM, and 6 histologically normal muscle biopsies underwent miRNA profiling to quantify the expression levels of ~800 miRNAs. For each type of IIM, we sought to identify miRNAs with significant differential expression. No statistically differentially expressed miRNAs were identified in IMNM muscle compared to normal muscle. In contrast, a differential expression of miRNAs was found in DM, AS, and IBM muscle (Figure 1a). In total, 36 miRNAs were differentially expressed compared to controls in one or more of these types of IIM muscle (Figure 1b). Using those miRNAs, we performed a Uniform Manifold Approximation and Projection (UMAP) analysis to visualize the distribution of the samples with respect to the different IIM types (Appendix A). We also analyzed the differences in expression between the IIM types for each of those miRNAs (Figure 1c and Appendix A), and compared the relative expression for each miRNA across the individual samples to confirm that the observed differences between the groups were not due to specific demographic or clinical features of the samples (Appendix A).

Compared to the controls, 11 miRNAs were identified as exclusively differentially expressed in DM, 7 as only differentially expressed in AS, and 9 as specifically upregulated in IBM. Several miRNAs were differentially expressed in more than one type of IIM compared to the control muscle. For example, miR-206 was upregulated in AS, DM, and IBM. In contrast, miR-135a-5p levels were significantly lower than controls in AS, DM, and IBM biopsies whereas miR-30e-3p expression was decreased only in DM and AS. Of note, compared to both controls and each of the other IIM types, miR-1246 was significantly upregulated in DM, whereas miR-299-5p and miR-150-5p were upregulated in IBM.

### 3.2. Correlation Analyses Suggest There Is a Coordinated Expression of Deregulated miRNAs and Regulation of Differentially Expressed Targets

We considered whether the expression of the differentially expressed miRNAs in IIM muscle biopsies might be coordinated. To investigate this, we calculated the correlation between the abundance of differentially expressed miRNAs across all samples. As expected, we found strong associations between the expression levels of miRNAs located in the same genomic cluster. Most evident was the correlation between miR-450a-5p, miR-424-5p, and miR-503-5p, each known to be transcribed from the H19X locus (Figure 2). We also found strong correlations between miRNAs clustered at the DLK1-DIO3 locus (miR-381-3p, miR-382-5p, miR-127-3p, miR-495-3p, miR-411-5p, miR-487b-3p, miR-543, and miR-299-5p).

In addition, we also observed positive correlations of expression between miRNAs that are not physically clustered together. The expression of these miRNAs might be coordinately regulated to modulate the expression of their targets. Using the multiMiR tool [20], which compiles 14 databases to collect miRNA targets that have been experimentally validated or computationally predicted, we identified putative targets of the differentially expressed miRNAs.

Then, we aimed to identify those mRNAs that might be targeted by the miRNAs in the context of each type of IIM. To do that, we combined our miRNA data with bulk mRNA expression data derived from the same samples (described previously [2,9,10]). We used the mRNA expression data to restrict the list of potential targets to those protein-coding genes that were dysregulated in the IIM type where the targeting miRNAs were differentially expressed. To further delimit the list of candidate targets, we selected those miRNA-mRNA pairs that were more likely to be associated in our samples. This selection was based on the presence of a significant correlation between the miRNA and the mRNA expression in the IIM type where both the mRNA and the miRNA are significantly deregulated. In the list, we included all the identified targets regardless of the positive or negative nature of the observed correlation with the miRNA. Thus, it could contain mRNAs whose differential expression might conceivably lead to changes in the expression of the targeting miRNAs as a compensatory mechanism to mitigate the deregulation of mRNA levels, although not yet effective or unsuccessful due to other potential deregulated processes. To provide an overview of the processes that the differentially expressed miRNAs might be modulating, we used these selected mRNA targets to perform a pathway enrichment analysis for each type of IIM using the Molecular Signatures Database (MSigDB) hallmark gene set collection [22]. This revealed an enrichment of target genes related to inflammation and cell cycle progression in both DM and AS (Appendix A).

When we repeated this analysis by including only those mRNAs with an inverse correlation with their targeting miRNAs in AS or DM (hypothesizing that these mRNAs could be differentially expressed in these types of IIM due to their negative regulation by the miRNAs) only the enrichment related to cell cycle progression was significant in AS (with an adjusted *p*-value of 0.0002 for both the G2M checkpoint and E2F targets gene sets). In DM, the most significant enrichments of the differentially expressed targets with a significant inverse correlation with the differentially expressed miRNAs were found for the oxidative phosphorylation (*p*-adjusted = 1.21 × 10^−8^) and for the interferon alpha and gamma response-related genes (*p*-adjusted = 7.55 × 10^−5^ and *p*-adjusted = 0.001, respectively) (Figure 3a).

The activation of interferon-signaling pathways, especially for type I interferon, is a well-described feature of DM [10,23]. Thus, we further explored the differentially expressed and inversely correlated miRNA-mRNA pairs in DM that involve targets included in the interferon-alpha response gene set. All those miRNA-mRNA pairs involved miR-30a-3p, miR-30e-3p, and miR-199b-5p. The negative correlations that we observed for miR-30a-3p and miR-30e-3p with their targets were stronger in the DM group, but also significant for most of the pairs in the non-DM samples. In contrast, the inverse correlation of miR-199b-5p with its targets was specific for DM (Figure 3b and Appendix A).

### 3.3. The Abundance of Differentially Expressed miRNAs and the Expression of Relevant Genes in Myositis Are Associated

Our findings support the possibility that the downregulation of miR-30a-3p, miR-30e-3p, and miR-199b-5p in DM contributes to the upregulated expression of interferon-regulated genes observed in this type of IIM. However, they do not exclude the possibility that interferon stimulation directly causes the downregulation of these miRNAs along with the upregulation of interferon-regulated genes. While these hypotheses are not mutually exclusive, we investigated the extent of the association of these miRNAs—and all the others identified as differentially expressed in one or more types of IIM in our study—with the interferon response. We calculated the correlation of their expression across all samples with the expression of mRNAs known to be regulated by interferon and observed a negative correlation of miR-30a-3p, miR-30e-3p, and miR-199b-5p with other genes that have not been described as their potential targets (Figure 4a). Additional miRNAs showed general negative or positive correlations with interferon-related mRNAs. Particularly strong were the positive correlations observed between most of the interferon-inducible genes with miR-361-3p or miR-222-3p levels, suggesting that there is an association of these miRNAs with the interferon response pathway.

We considered whether the assessment of correlations between miRNA and mRNA abundance could also yield insights regarding other transcriptomic pathways relevant to IIM. Therefore, we extended our analysis to evaluate the correlation between the expression of miRNAs and genes with known associations with IIM pathology. This revealed that the expression of most of the common downregulated miRNAs was positively correlated with the expression of genes encoding mature-muscle structural proteins and negatively correlated with those implicated in muscle regeneration. The opposite was the case of the commonly upregulated miRNAs and the IBM-upregulated miRNAs with a similar profile, which showed a generally stronger positive correlation with some genes involved in muscle regeneration (Figure 4b).

With regard to levels of miRNA and their associations with genes expressed in immune cells, the strongest correlations were between miR-146b-5p and macrophage genes (CD14 and CD68) and CD4. However, a general moderate positive correlation was noticed for many of the common and IBM-specific upregulated miRNAs with most of the genes associated with inflammatory cells. The opposite was observed with the commonly downregulated genes, which had weak negative correlations, especially for levels of the macrophage genes CD14 and CD68 (Figure 4c). Overall, these observations revealed a probable association of differentially expressed miRNAs with muscle damage, muscle regeneration, and inflammatory cell infiltrates.

## 4. Discussion

In this study, we used the NanoString miRNA assay to evaluate the expression profiles of miRNAs in muscle biopsies from patients with AS, DM, IBM, and IMNM. This revealed several miRNAs whose expression was only significantly altered in one type of IIM. For example, compared to both controls and to each of the other IIM types, miR-1246 was only upregulated in DM, whereas miR-299-5p and miR-150-5p were only upregulated in IBM. We also identified miRNAs that were commonly deregulated in more than one type of IIM. For example, expression levels of miR-206, miR-21-5p, miR-424-5p, miR-450a-5p, miR-503-5p, miR-133b, and miR-222-3p were upregulated, whereas miR-135a-5p and miR-30e-3p were downregulated in at least two types of IIM.

Many of the commonly deregulated miRNAs have already established associations with muscle physiology and muscle-related disorders [4,5,24,25]. Published research on muscle-related miRNAs in IIM has paid special attention to the miRNAs known as classical myomiRs. Studies on blood and muscle from IIM patients resulted in contradictory findings about the specific myomiRs showing significant deregulation and about the direction of the altered expression [26,27,28,29,30,31]. Those discrepancies might be due to differences in the stage of the disease, the treatment of the patients, or the techniques used to assess the expression. In our samples, the only differentially expressed myomiRs were miR-206 and miR-133b. They were both upregulated in most of the IIM muscle samples relative to histologically normal muscle, although the expression pattern of miR-133b showed a more pronounced upregulation in DM. Regarding other common deregulated miRNAs, our findings support existing evidence on the upregulation of miR-21 [27,31,32,33] and miR-146b [27,28,31,32,34,35,36], as well as on the downregulation of miR-135a [31] in IIM. Our analysis also extends the knowledge on the changes in expression of other miRNAs identified in our study, for which regulation in IIM has been inconsistent (e.g., miR-504 [31,37]) or barely reported (e.g., miR-503-5p [34], miR-424-5p [27], and miR-450a-5p).

Most of the IBM-specific differentially expressed miRNAs belong to the miRNA cluster at the DLK1-DIO3 locus. It comprises miR-299-5p, which was significantly upregulated in IBM relative to NT and to each of the other IIM types, and miR-381-3p, miR-382-5p, miR-127-3p, miR-495-3p, miR-411-5p, miR-487b-3p, and miR-543, with similar expression profiles. Previous studies have established associations between miRNAs located at this locus and muscle regeneration [38,39], which is in agreement with the positive correlation that we observed with markers of this process. However, few contradictory observations have been reported about the expression of some of the miRNAs of this cluster in different types of IIM [27,34,37,40]. Intriguingly, the cluster is upstream of the IGH locus on chromosome 14q32 and the upregulation of immunoglobulins has been described as a feature of IBM [41,42]; however, we could not find an evident association between these changes based on the correlations between the expression of the miRNAs in this cluster and the immunoglobulin genes at the neighboring locus.

The other miRNA that we identified as upregulated in IBM relative to NT and to each of the other IIM types is miR-150-5p. The specificity of this change is in accordance with the upregulation that has been observed in IBM and not in other types of IIM muscle [27], which might imply the potential use of miR-150-5p as a biomarker of IBM in muscle. This miRNA is involved in B- and T-cell differentiation [43,44]. Therefore, although correlations of miR-150-5p with common markers of these or other immune cell types were not apparent in our study, it is possible that its upregulation is related to the presence of certain subpopulations of differentiated B- and T-cells in the muscle of IBM patients [45,46].

In DM, miR-1246 was significantly upregulated compared to control and other IIM types. This miRNA has been described in the literature as a biomarker of anti-MDA5^+^ DM samples with interstitial lung disease or DM samples in general [30,47]. However, other IIM types were not included in those studies. In plasma exosomes from DM patients, miR-1246 was described to be downregulated by treatment, although the decrease did not reach statistical significance in a validation experiment [30]. While differences in treatment could explain the distinct expression levels, our results do not support any difference in miR-1246 between patients who received or did not receive any treatment. In addition, an upregulation of miR-1246 has been described when muscle cells were incubated with exosomes from DM patients or with interferon [30], but we could not find substantial correlations of miR-1246 with genes of the interferon response, or the other mRNAs that we studied in our work.

Distinct interferon-stimulated gene expression signatures have been described in the different types of IIM [10]. For example, DM has a very prominent type I interferon gene signature compared to the other types of IIM. Interestingly, we found an association between the expression of some miRNAs with the expression of interferon-stimulated genes. Supporting the relationship of the altered expression of these miRNAs with the interferon response, it has been shown that interferon upregulates the expression of miR-128, miR-146, and miR-222-3p while downregulating the expression of miR-299-3p, miR-199b-5p, miR-518b, and the exact isomiR of miR-221-3p assessed in our panel [48,49]. As miRNAs modulate gene expression, it is possible that differentially expressed miRNAs could help shape the interferon gene signatures found in the different types of IIM.

We identified strong inverse correlations between the abundance of miR-199b-5p, miR-30a-3p, and miR-30e-3p and the expression of mRNAs identified as their targets and that are upregulated by interferon. While miR-199b-5p levels are lower after interferon treatment, the downregulation of the other two has not been observed in the same experimental conditions [49], and there are contradicting reports about IFN-induced changes in the regulation of miRNAs of the miR-30 family in different cell types [48,49]. In human skeletal muscle myoblasts, non-significant changes in miR-30a-3p and miR-30e-3p after stimulation with interferon were observed by Gao et al. [50]. They also incubated myoblasts with serum from IIM patients, which led to a non-significant upregulation of miR-30e-3p, whereas a trend towards miR-30e-3p downregulation was observed in the muscle from IIM patients [50].

The downregulation of miR-30a-3p, miR-30e-3p, and other miR-30 miRNAs in muscle tissue from IIM patients has been reported in other studies [27,31,47,50]. Interestingly, miR-30 miRNAs participate in myogenic differentiation and have been proposed as biomarkers of muscle homeostasis [51,52]. In addition, they have been shown to regulate the interferon response [53,54,55,56]. We speculate that the lower expression of miR-30a-3p and miR-30e-3p could contribute to heightening the interferon signature observed in distinct IIM types, such as DM. In that case, manipulating their expression could be a potential approach to dampen the interferon response. Additional experimental evidence will be needed to verify this hypothesis.

This study has several limitations. First, while including a relatively large number of samples of each IIM-type, some less common MSAs (e.g., non-anti-Jo1 AS) were not present in this cohort, and this study could still have been underpowered to detect the deregulation of some miRNAs in certain types of IIM. Second, some of the miRNAs identified in this study have been identified in prior studies of IIM muscle. However, most of these studies did not include samples from patients with all four major IIM types (DM, AS, IMNM, and IBM). Third, some of our results contradict the findings of prior reports (e.g., the absence of differential expression of other myomiRs in addition to miR-206 and miR-133b). This may be explained by differences in the severity of the disease, treatment of the patients, or any other feature with respect to the individuals and the biopsies. Finally, the heterogeneity of the samples with regard to demographic and other clinical features, while representative of each type of IIM, should be considered when interpreting the results.

## 5. Conclusions

Our analysis of miRNA expression has allowed us to identify miRNAs that are deregulated in specific types of IIM muscle as well as those that are upregulated or downregulated in more than one type of IIM. Furthermore, by combining miRNA data with mRNA expression data from the same samples, we have uncovered miRNA-mRNA relationships that could be important for IIM pathogenesis. However, additional studies will be needed to determine the role of the miRNAs in the onset and progression of IIM and to explore the value of their aberrant expression as biomarkers of the disease or as the basis for the development of new treatments.

## Figures and Tables

**Figure 1 cells-12-02198-f001:**
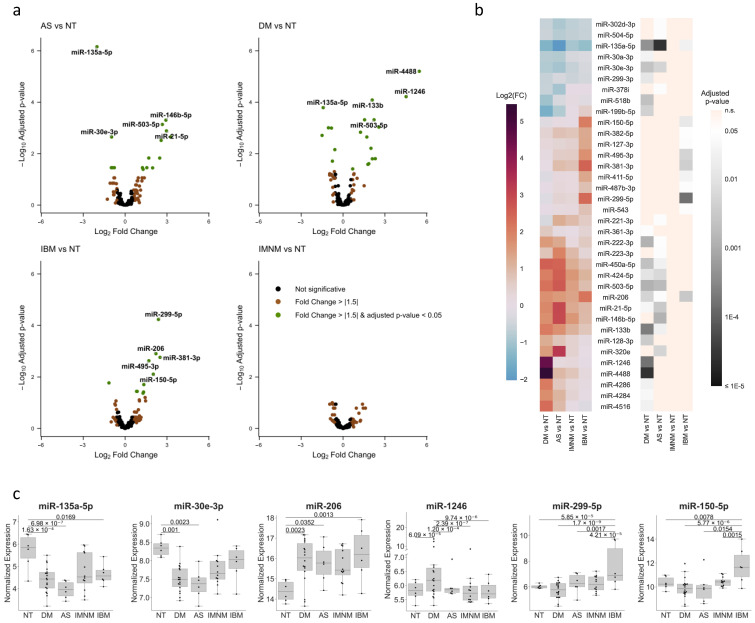
Expression profiles of differentially expressed miRNAs in each type of IIM. (**a**) Volcano plot representation of the differential expression analysis of each type of IIM (AS, DM, IBM, or IMNM) relative to histologically normal tissue (NT). Labels indicate the five differentially expressed miRNAs with the lowest *p*-adjusted value for each IIM type. (**b**) Heatmap representations including the fold change (**left**) and adjusted *p*-values (**right**) resulting from the differential expression analysis for each type of IIM relative to NT, selecting all the miRNAs identified as differentially expressed in at least one IIM type compared to NT. (**c**) Graphs represent log-scaled normalized expression levels of differentially expressed miRNAs that are representative of common IIM or IIM type-specific changes. The expression profile of the rest of the differentially expressed miRNAs identified can be found in Appendix A. Note that, for the miRNAs whose expression was exceptionally high in a few samples, the y-axis has been broken for a better representation of the expression in the rest of the samples.

**Figure 2 cells-12-02198-f002:**
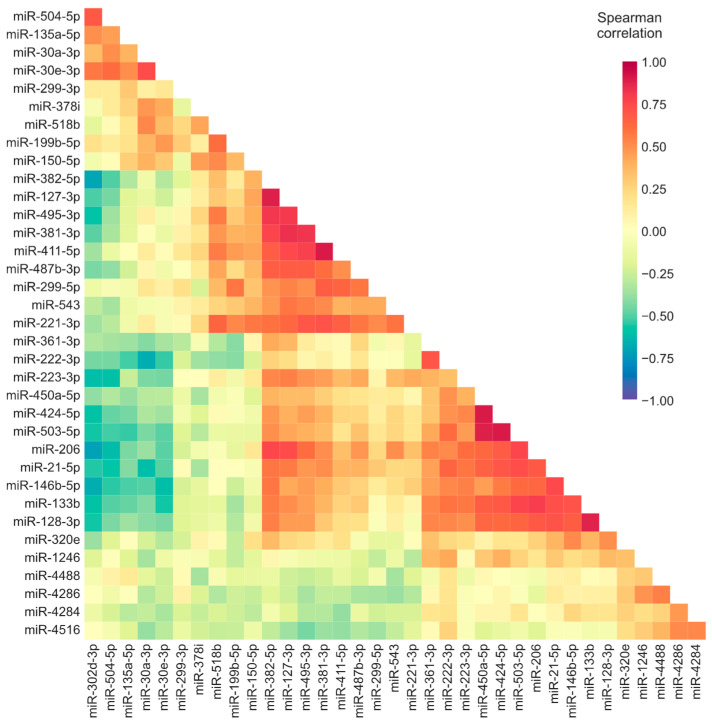
Correlation of the expression of differentially expressed miRNAs. Heatmap representation of the Spearman’s rank correlation coefficients between the expression levels of the miRNAs that were differentially expressed in one or more types of IIM muscle compared to histologically normal muscle.

**Figure 3 cells-12-02198-f003:**
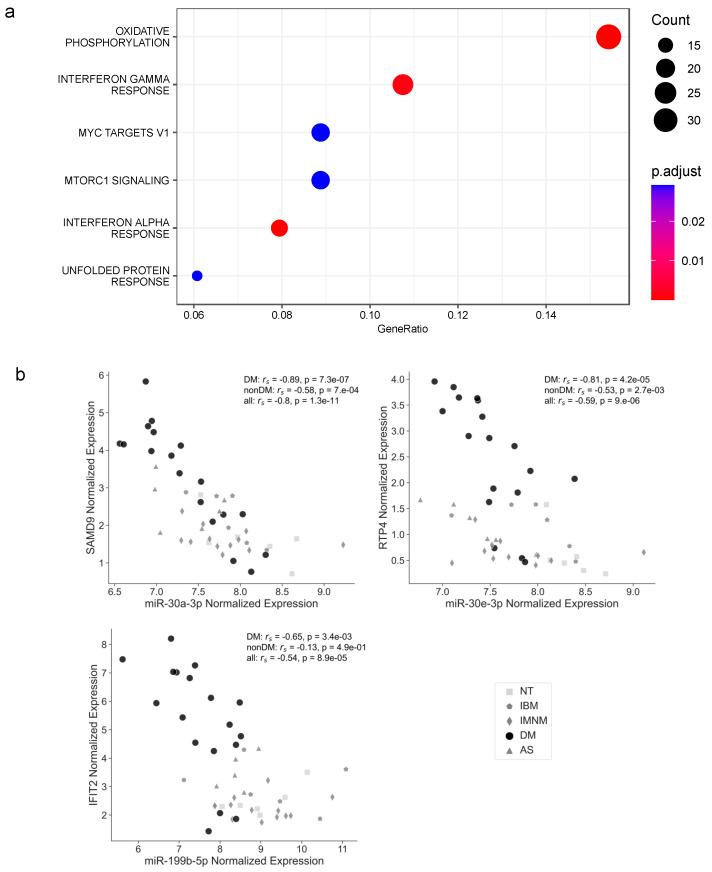
Differentially expressed and negatively correlated miRNA-mRNA pairs in DM involve mRNAs enriched in the interferon pathways. (**a**) Dot plot shows the pathway enrichment of DM differentially expressed mRNAs that are negatively correlated with miRNAs identified also as DM differentially expressed. (**b**) Scatterplots represent the correlation of miR-30a-3p, miR-30e-3p, and miR-199b-5p with their most negatively correlated target in DM that is also differentially expressed in this IIM type and associated with the interferon-alpha response. The plots of the rest of the DM negatively correlated miRNA-mRNA pairs involving mRNAs included in the interferon-alpha response gene set can be visualized in Appendix A. Spearman’s rank correlation (r_s_) and *p* values of the correlation across DM samples, all samples excluding DM samples (named nonDM for short), or the total of samples (all), are shown.

**Figure 4 cells-12-02198-f004:**
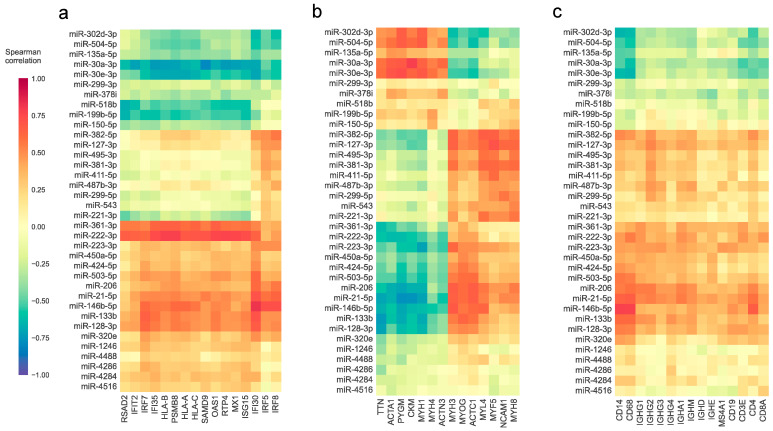
Correlation between the expression of differentially expressed miRNAs and relevant genes in IIM. Heatmap representation of the Spearman’s rank correlation coefficients between the expression levels of miRNAs that were differentially expressed in one or more types of IIM and genes that are stimulated by interferon (**a**), genes that are abundant in mature or differentiating muscle (**b**), or genes whose expression is enriched in immune cells (**c**) across all samples.

## Data Availability

The data presented in this study are available upon request from the corresponding author.

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
