# Peer review of "Identification of Unique microRNA Profiles in Different Types of Idiopathic Inflammatory Myopathy"

_cells, 2023, doi:10.3390/cells12172198_

Round 1

Reviewer 1 Report

The work presented here under the title Identification of unique microRNA profiles in different types of idiopathic inflammatory myopathy written by Munoz-Braceras et al is a very nice work. The team presents here different molecular fingerprint among different inflammatory myopathy. Those fingerprint were identified by quantification of miRNA using Nanostring ncounter system.

The whole article is very well written and is a pleasure to read. The overall design of the study is well described.

But it becomes apparent that there may be a potential concern with the study's design that was not discussed. Reading "Supplementary table 1 . Samples included in the study. ",  there seems to be a lack of homogeneity with respect to gender, ethnic origin, and the biopsy site. As it is possible that all the features could induce variation.  Without accounting for these differences, the study's findings could be biased or limited in their applicability to broader contexts. To address these concerns, it is recommended that the study authors thoroughly justify and explain their decisions regarding participant selection, taking into account the potential impact of gender, ethnic diversity, and biopsy site on the study outcomes. Furthermore, a  statistical analysis should be conducted to assess whether any observed effects are consistent across different subgroups defined by gender, ethnic background, and biopsy location.

In addition, for enhanced differentiation among the various groups, the inclusion of a t-distributed Stochastic Neighbor Embedding (t-SNE) or Uniform Manifold Approximation and Projection (UMAP) analysis of the complete dataset could serve as an instructive visual aid for readers.

Reviewer 2 Report

The authors present a study on miRNAs in IIM. Briefly, they analysed 35 muscle biopsies of IIM patients and demonstrate specific signatures in each entity.

The study of miRNAs in IIM has a clear novelty factor and is of current interest to the field of myositis.

Before publication, the paper would benefit from adressing the following comments:

- In line 68, the authors state that "The six control samples had been obtained from histologically 78 normal muscle biopsies from non-IIM patients.". Please expand why these biopsies were considered to be "non-IIM". What symptoms did they present? What was the CK? Please provide clinical information for the entire cohort in respect to treatment and antibody status for IBM (were alls IBM patients seronegative?). Please also provide representative IHC stainings for these patients and for the IIM patients as supplement.

- The legend in Figure 1b is not readable.

- A graphical representation for the dataset would improve the scope. Did the authors try to perform dimensional reduction using a PCA or UMAP? If the patients do not cluster using the entire dataset, it might be adviced to include only significantly changed miRNAs for a PCA and clearly state so in the methods. Similarily, a heatmap of differently expressed miRNAs might be beneficial.

- The gene enrichment analysis for DM is fitting to the current knowledge on pathophysiology. What about IMNM, AS and IBM? The authors are advised to also include these entities.

- The authors should also highlight in the discussion that all AS patients were Jo1.

Overall, this is an interesting and well presented paper.

Round 2

Reviewer 1 Report

Dear authors, 

I agree with all the answer and proposition made by your team. 

I am sure it will help the reader to better appreciate and understand the article.

But it would be very important to add a tag for each biopsy in the S1 UMAP plot which could link to the S3 heat map. Otherwise the reader won't be capable to appreciate the UMAP . And it is only a line of code, it won't be a hard work )

best regards
